# The role of grammar in transition-probabilities of subsequent words in English text

Rudolf Hanel[1,2☯], Stefan Thurner[1,2,3,4☯] *

**1** Section for Science of Complex Systems, Medical University of Vienna, Vienna, Austria, **2** Complexity Science Hub Vienna, Vienna, Austria, **3** Santa Fe Institute, Santa Fe, NM, United States of America, **4** IIASA, Laxenburg, Austria

☯ These authors contributed equally to this work.
* stefan.thurner@muw.ac.at

## Abstract

Sentence formation is a highly structured, history-dependent, and sample-space reducing (SSR) process. While the first word in a sentence can be chosen from the entire vocabulary, typically, the freedom of choosing subsequent words gets more and more constrained by grammar and context, as the sentence progresses. This sample-space reducing property offers a natural explanation of Zipf's law in word frequencies, however, it fails to capture the structure of the word-to-word transition probability matrices of English text. Here we adopt the view that grammatical constraints (such as subject–predicate–object) locally re-order the word order in sentences that are sampled by the word generation process. We demonstrate that superimposing grammatical structure–as a local word re-ordering (permutation) process–on a sample-space reducing word generation process is sufficient to explain both, word frequencies and word-to-word transition probabilities. We compare the performance of the grammatically ordered SSR model in reproducing several test statistics of real texts with other text generation models, such as the Bernoulli model, the Simon model, and the random typewriting model.

**Data Availability Statement:** All data entering the analysis presented in the paper is available in Project Gutenberg: https://www.gutenberg.org.

**Funding:** ST P29032 and P29252 Austrian Science Fund FWF https://www.fwf.ac.at The funders had

## Introduction

After almost a century of work, understanding statistical regularities in language is still work in progress. Maybe the most striking statistical feature is that rank ordered distributions of word frequencies follow an approximate power law,

$$f(r) \sim r^{-\alpha} \quad , \tag{1}$$

where $r$ is the rank assigned to every word in a given text; the most frequent word has rank one, the second most frequent has rank two, etc. For most word-based texts, one finds $\alpha \sim 1$, independent of language, genre, and time of writing. This "universal" feature is called Zipf's law [1]. Fig 1 shows the rank distribution of words in the first text "Five little peppers and how

no role in study design, data collection and analysis, decision to publish, or preparation of the manuscript.

**Competing interests:** The authors have declared that no competing interests exist.

**Fig 1. Rank ordered frequency distribution of the most frequent 1000 words in the novel "Five little peppers and how they grew", by M. Sidney (green).** The result from the SSR model (gray) and its rank ordered frequency distribution (red) show an exact Zipf's law, and are both invariant under grammatical re-shuffling. The SSR sequence has been produced for approximately $W = 5000$ words, and $N = 50,000$ samples (gray).

they grew" (green), the first novel in a book series by M. Sidney published between 1881 and 1916.

There are several ways to understand Zipf's law through entirely different mechanisms. Zipf's first qualitative explanation of the phenomenon was based on communication "efforts" of sender and receiver [1], an idea that was later expressed in an information-theoretic framework [2–5]. The first quantitative linguistic model by H. Simon features the idea of preferential attachment, i.e., words are added to the text with a probability that is proportional to their previous appearance in the text. New words are added at a low, constant rate [6]. Zipf's law follows immediately from those two assumptions. Preferential attachment models were later refined [7], relating Zipf's law to Heap's law [8]. The conceptually simplest way to understand Zipf's law are random typewriting models, where words and texts are created by randomly typing on a typewriter [9–11]. Yet another route to Zipf's law was introduced on the basis of sample-space reducing (SSR) processes [12], which successively reduce their sample-space (range of potential outcomes) as they unfold [13]. SSR processes generically produce power laws, and Zipf's law in particular [14]. The sample-space reducing process based intuition of how sentences form is that, as we express what we want to say into a stream of words, contextual, grammatical, and maybe other types of requirements ad constraints to our word choices as the narrative unfolds. While contextual constraints typically act globally, on all scales or levels of the narrative, grammatical rules, conventions, or habits constrain word order locally at the level of phrases, sentences and perhaps paragraphs. At the global level the intention underlying a narrative induces a contextual ordering of the set of available words towards core elements of the narrative. This ordering can be captured by a SSR type of word selection process that,

conditional on the intended content and context of the narrative, guides and directs word choices at the local level of sentences as well. For instance: one can pick any word to start a sentence. Once the first word is chosen, grammar and context constrain the possibilities for choosing the second word. The choice of the second word further constrains the possibilities for the third word, and so on, as the sentence or paragraph, in terms of word choices, converges towards its intended content. The sample-space of possible words generally reduces as sentences form. In this view of sentence formation, grammar, but mainly context, constrain the choice of words later in the sentence; we therefore perceive text generation as a SSR process, and Zipf's law must follow. The existence of grammatical and contextual constraints allow us–at the receiving part of a communication–to complete sentences in advance, and to anticipate words that will appear later. This (at least partially) ordered hierarchical structure *guides* sentence formation and allows a receiver to robustly decode messages [15]. For instance, if the context is "Electrodynamics" we expect the narrative to return to a few notions, the protagonists of the narrative, which in this case would include notions such as "charges", "electrons" or the "electro-magnetic field". One can view those central notions being embedded in contexts of various specificity the narrative spans, inducing a more or less hierarchical order on the set of notions we require in order to speak about the topic. The SSR structure of word selection therefore becomes plausible by realizing that any story line needs to connect the "protagonists" with their respective context. If we think of the hierarchically structured context as a network of words, then the process of directly or indirectly linking any chosen word to a given protagonist is comparable to a *targeted diffusion* process on the word network, where protagonists (and other central words) are the targets in this diffusion process. Targeted diffusion is an example of SSR processes, and generically leads to power laws in visiting frequencies [16].

Note that the range of constraints governing realistic word selection process in narratives is not necessarily limited to the scale of sentences but may extend over whole paragraphs or sections, a complexity that simple generative statistical models of text will hardly capture. What such simple models provide us with is a way to test the extent to which very basic percepts of language formation, such as the intuition we sketched above, or the original intuition of Zipf, allow us to understand similarly basic statistical features of text, for instance Zipf's law.

To understand Zipf's law of word frequencies, however, is not the end of the story. There are also the frequencies of word $i$ to follow word $j$ to consider. The word frequency distribution is the marginal distribution of the joint word distribution with conditional word transition probabilities, $p(i|j)$, the probability to produce word $i$, given that the previous word was $j$; see Fig 2. We will refer to $p(i|j)$ also as the word transition matrix $A$. Fig 2A shows the transition matrix for "Five Little Peppers". Pure SSR processes show a triangular structure in the transition probabilities, see Fig 2B, which obviously does not match the empirically observed transition matrices of real texts. Empirical transition probabilities look similar to transition matrices that correspond to word sequences that have been sampled independently from Zipf distribution, i.e. the (typically) observed marginal empirical word distribution function (Bernoulli model), Fig 2C. Empirical transition probabilities look as if they were random (in this sense), even though the generation of real texts is obviously a highly structured generative processes.

In this paper we assume that the formation of word sequences is a combination of two processes: The *word selection process*, selects the words that are needed to encode a narrative or to convey a coherent message or meaning. The other process is *grammar*, which, at the sentence level, brings the selected words into a specific order. We assume that the word selection process is of SSR type, the context created by the generated words restricts the usage of other words as the sentence progresses. Grammatical ordering can be thought of as a post-processing of the word stream generated by the word selection process. It establishes a local word

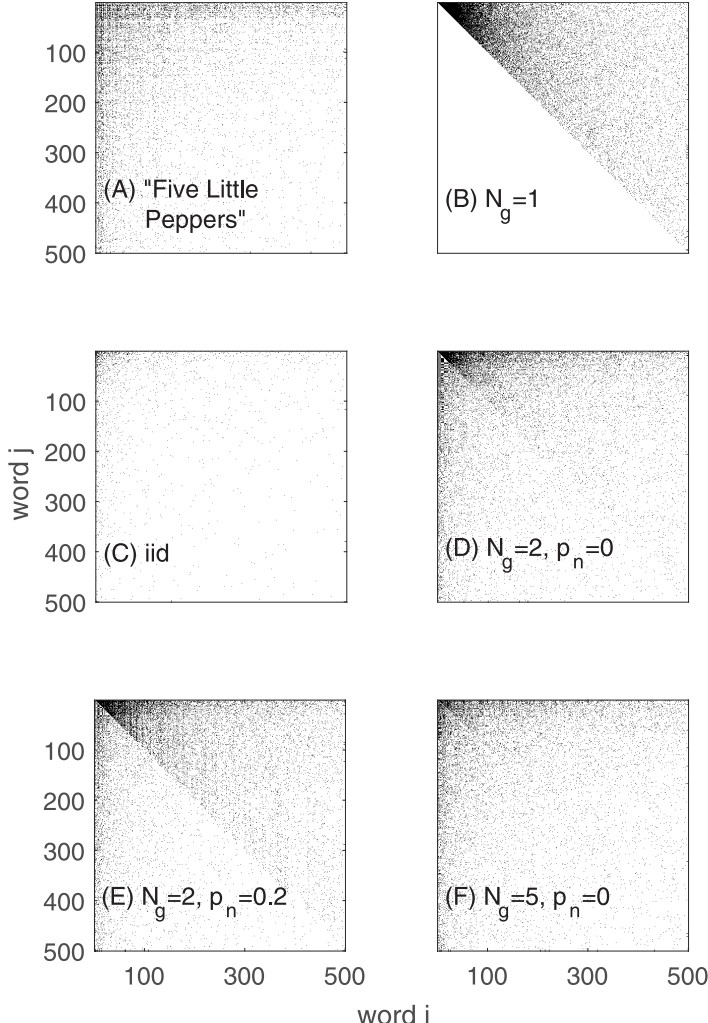

**Fig 2. Structure of word transitions.** (a) Word transition matrix $A_{ij}$ (first index is x-axis, second is y-axis) of the text "Five little peppers and how they grew". $A$ is plotted in such a way that a point in a plot indicates that word $i$ followed word $j$ at least once in the text. Words are ordered along the axis according to their frequency of appearance. (b) Transition matrix of a pure SSR process. States are ordered according to their natural index $i = 1, \cdots, W$. The triangular structure is visible. (c) The Bernoulli model for the same novel shows the matrix for words sampled independently from the marginal word frequency distribution of the novel. (d-f) Transition matrices for goSSR processes for $N_g = 2$ and 5 grammatical categories. In (d) and (e) we compare the situation with different probabilities for neutral elements $p_n = 0$ and $p_n = 0.2$. As $N_g$ increases, the goSSR transition matrices begin to resemble those of actual text. All transition probabilities are computed for a lexicon containing roughly 4, 800 words. Samples consist of approximately 88, 000 word transitions and 5, 200 sentences.

order, which differs from the order the words were generated, but is in line with grammatical expectations, such as the subject–predicate–object (SPO) order that is typically used in (very simple) English sentences, such as "Peter watches clouds". In other words, grammar locally scrambles (permutes) the order of the word selection stream, while keeping the underlying word frequency distribution invariant.

We will show here that if the word selection process is a pure SSR process, and if generic grammatical rules locally re-shuffle the word order of the word selection stream, the resulting word transition probabilities have statistical properties that closely resemble the empirically observed ones. We will see that grammar strongly masks the triangular structure introduced

by the considered word selection process. This implies that empirical word transition probabilities provide us with only very limited information about the underlying word selection process that generates an information-carrying narrative. In particular, with a simple SSR model of word generation we demonstrate how imposing grammatical rules of variable strength changes the transition probabilities from structured (triangular, Fig 2B) to seemingly unstructured, Fig 2C. Note that the grammatical ordering process would locally transform word streams from any adequate (Zipf law producing) word selection process in a similar way by destroying information about of the word selection process that one otherwise could simply infer from the word transition matrix.

We first discuss SSR processes and their frequency distributions and then introduce a modification, where the local word order of SSR sequences is permuted to conform with a grammatical word order, such as for instance the simple SPO scheme already mentioned above. To implement a "grammatically ordered" SSR (goSSR) process we assign grammatical labels $c \in \{1, \cdots, N_g\}$ to all words contained in the lexicon of used words. In the model with $N_g = 3$ we could think of $S \equiv 1$, $P \equiv 2$, and $O \equiv 3$ simply as the three labels of the toy grammar we introduce below. The labels determine the *local* order in which words appear in a sentence. We compute statistical properties of natural language in an English text corpus [17] and compare them with those obtained from goSSR processes. We find that typically three to five grammatical labels suffice to produce realistic results. We finally compare the Bernoulli-, Simon-, the random typewriting-, and the goSSR-models for six texts in English language, with respect to two statistical measures that also serve as test-statistics for hypotheses testing. As a null-hypothesis we assume that sequences have been generated by a Bernoulli process. This provides us with a first quantitative understanding of how informative these models are with respect to actual text generation.

Some final introductory remarks. Note that notions of explanatory adequacy of models such as the proposed model, the Simon model, or random typewriting model, are based on considering statistical features we select for comparison and are quantitatively used in terms of test statistics. Such a comparison can only tell us how models perform with respect to those selected features. Similarity and adequacy are obviously context sensitive notions that depend on the set of considered features. Obviously we could produce a plethora of test-statistics where all the simple generative models of text we consider in this paper fail to resemble the respective texts they mimic. They are models after all. It is therefore important to stress that the aim of this paper is to demonstrate that the two observable features, Zipf's law and random-looking word transition probabilities, can be explained at the same time by considering a word selection process that generates Zipf's law and a grammatical ordering process that locally, at the sentence level, re-shuffles the word order of the selected word stream. The SSR process as word selection process is plausible but what is crucial is that the word selection process already produces Zipf's law for rank ordered word frequency distributions. That is, the local grammatical word ordering process becomes responsible for the observable shape of empirical word-transition frequencies and could also be superimposed on other generative models producing Zipf's law, for instance the Simon model. A detailed analysis of word selection process would therefore require a computationally very expensive analysis of a large body of texts where possible dependencies of results on text length, average sentence length and other measurable features of written texts need to be carefully controlled and studied. However, the message of this paper is much simpler. Two highly structured processes, when acting upon one another can, at a basic statistical level, resemble a Bernoulli (iid) sampling process. That is, grammar can, at the level of word transition probabilities, make any adequate word selection process, even if this process is highly structured such as SSR processes, resemble independent sampling from Zipf's distribution.

We will therefore merely exemplify the adequacy and robustness of this intuition using six texts that have been selected to cover a range of texts lengths and types (including for instance a translation of Grimm's tales, from their German original into English). A systematic study of generative statistical models superimposed with grammatical reordering processes, on the basis of a large text corpus, goes far beyond the scope of this paper and would possibly over-state the concreteness of the hypothesis we discuss here.

## Grammatically ordered SSR model

We first discuss the *sample-space reducing* process for sentence formation and then augment it with a toy grammar. SSR processes are characterized by $W$ linearly ordered states. For example think of a staircase. The lowest stair is state 1, the next step is state 2, and so on; at the top of the staircase we have state $W$. Imagine a ball bouncing downward this staircase with random jump sizes. We begin at state $W$. The ball can jump to any of the $W - 1$ lower states; say it jumps to state $x_1$ (subscript indicates the first jump). Obviously, $1 \leq x_1 \leq W - 1$. Again, the next state, $x_2$, can only be a lower state, $1 \leq x_2 < x_1$. After a sequence of $n - 1$ visits to states $x_1$, ..., $x_{n-1}$, the ball reaches the bottom of the staircase, $x_n = 1$. At this state the process needs to be restarted, which means lifting the ball to any randomly chosen state, $1 < x_{n+1} \leq W$. If the process gets restarted multiple times, the visiting distribution of the process to states $i$ appears to be exactly Zipf's law,

$$p_i = \frac{1}{Z}\frac{1}{i} \quad , \tag{2}$$

where $Z$ is a normalization constant [13].

In real texts the SSR word selection sequences may in principle not be strictly bound to the scale of sentences and could extend to larger lingual units such as paragraphs. One could think of modelling this by sampling a number of sentences that together form a single SSR sequence, for instance. However, since this would not qualitatively change the effect we want to demon-strate here–that grammar can locally scramble selected word sequences to conform with gram-matical rules and in doing so may make the grammatically ordered word sequence look as if it were sampled by a Bernoulli process–we refrain from considering this possibility and for rea-sons of model simplicity assume that each sentence starts a new SSR word selection sequence. That is, sentence formation, can be seen as an SSR process [12]. Words are not randomly drawn from the sample-space of all possible words (lexicon) but are used in context and gram-matical order. The fact that words in a sentence restrict the usage of consecutive words, gener-ates a SSR process. Imagine the first word group in a sentence is randomly drawn from the entire lexicon with $W$ words (states), say "The", an article that expects a noun, say "wolf". As soon as this is decided, the first word group can either continue and elaborate "The wolf" fur-ther, e.g. by stating something like "The wolf with the reddish fur". Note that outside any speci-fied context we can of course think of a plethora of ways to elaborate "The wolf". In a narrative about a particular wolf-pack the particular context constrains this arbitrariness and the same wolf would have to be referred to in similar ways again and again. After the first word group ends the second group begins and typically needs to be or include a verb (grammatical restric-tion), which in turn has to create a meaningful context (context dependent restriction) with the first group "The wolf". The selection of a verb such as "is typewriting", for instance, will be very unlikely in non-fiction texts. Conversely, if the word selection process should for instance start with the group "is typewriting" then it is very unlikely to chose "The wolf" as a subject. As sentences progress, typically more and more constraints occur and together restrict the subse-quent words that can be realistically selected. Once the final word of a sentence is selected, i.e.

once the SSR selection process halts or is stopped, a new SSR sequence, i.e. the next *word selection stream*, starts. Sample-space reduction in text formation is necessary to robustly convey meaningful information, a fact that is for example exploited by text-completion apps. While rank ordered word frequency distribution functions can be explained by the hypothesis that sentence and text formation follows SSR processes, word transition probabilities can not. Transition probabilities for pure SSR processes are triangular, see Fig 2B, and do not resemble those of actual text, Fig 2A.

We assume that the word selection process for a given narrative can be approximated by a SSR process that models the context-dependent constraints only. Grammar enters as a process that enforces locally "correct" word order. Effectively, it locally re-assembles the word sequences generated by the SSR word selection process, and destroys local correlations of word occurrences as generated by the SSR sequence. We now augment SSR models with a "grammar" that determines the local ordering of words.

To implement a toy-grammar, assume that there exist $N_g$ grammatical labels that are associated to words. Every word, $i = 1, \cdots, W$ in the lexicon, carries one of the $N_g$ distinct labels, $L(i) \in \{1, 2, \cdots, N_g\}$. For example, if $N_g = 3$, the three labels could represent $S \equiv 1$, $P \equiv 2$, and $O \equiv 3$. For simplicity we assume that each label appears with approximately the same frequency in the lexicon. In addition there exists a grammatical label $L(i) = 0$ that we call the "neutral" label. Neutral elements are combined with the next non-neutral word (with label $L > 0$) that follows in the text. This word complex is then treated as a single word with label $L$. The probability of finding neutral words in the text is denoted by $p_n$. We can now formulate the "grammar rules":

- (i) Words must follow a strict repeating pattern of grammatical labels, $1 \rightarrow 2 \rightarrow 3 \rightarrow \cdots \rightarrow N_g \rightarrow 1 \rightarrow 2 \rightarrow \cdots$. For example, if $N_g = 3$ ($S \equiv 1$, $P \equiv 2$, and $O \equiv 3$) we will produce sentences with a sequence of labels: $\cdots \rightarrow S \rightarrow P \rightarrow O \rightarrow S \rightarrow P \rightarrow O \rightarrow \cdots$.

- (ii) Missing labels are skipped. If $N_g = 4$ and if label 3 is not present in a particular sentence, but labels 1, 2 and 4 are, we order words according to existing labels: $1 \rightarrow 2 \rightarrow 4$.

- (iii) Neutral elements (label 0) do not change their relative position to the next non-neutral element following in the sentence. Neutral elements together with their adjacent non-neutral element form a complex that in grammatical reordering is treated as a single word. This means that the local SSR sequence order of grammatically neutral fragments is untouched by the grammatical ordering process.

For example, a SSR sequence of words (states) 260, 120, 76, 45, 13, 12, 7, 1 is generated in a $N_g = 3$ grammar with the corresponding grammatical label sequence: 3, 1, 0, 0, 2, 3, 0, 1. The grammar-ordered label sequence is 1, 0, 0, 2, 3, 0, 1, 3, and the grammar-ordered state sequence becomes 120, 76, 45, 13, 260, 7, 1, 12.

The goSSR model of a given text is the following. Determine the number $W$ of distinct words in a text. Produce a random map $L$ that associates a label $0 \leq L(i) \leq N_g$ with each word $i = 1, 2, \ldots, W$. The neutral label $L(i) = 0$ gets sampled with probability $p_n$, all other labels with probability $(1 - p_n)/N_g$. For each sentence in the text determine its length (in words) and generate a random SSR sequence of the same length. Then re-order this SSR sequence according to the grammar rules to get a grammatically ordered sentence. We call this sampled new text, the *goSSR model* of some original text. It contains as many words and sentences as the original text. Each sentence is a goSSR sequence.

For comparison, we consider three other models of text generation that yield power law distributed rank frequencies. The simplest is produced by independently sampling words from the word frequency distribution of the original text, which we refer to as the *Bernoulli model* of

the original text. The others are the Simon model, and the random typewriting model of text, see Methods.

## Results

We consider six English texts from the Gutenberg text corpora, [17]:

1. *The Two Captains* (abbreviated as 2cpns or Two Captains), by F. de La Motte-Fouque, with a length of roughly 18,000 words, 600 sentences, and a vocabulary of roughly 3,000 words.

2. *Five little peppers and how they grew* (5lpep or Five little peppers), the first in a series of books by M. Sidney, with a length of approximately 88,000 words, 5,200 sentences, and 4,800 distinct words,

3. *Collected tales by the Brothers Grimm* (grimm or Grimm), a collection of tales, with 180,000 words, 4,500 sentences, and about 4,800 words.

4. *Moby Dick; or the whale* (moby or Moby Dick), by Herman Melville, with about 250,000 words, 10,000 sentences, and 17,000 distinct words,

5. *The age of invention, a chronicle of mechanical conquest* (aoinvnt or The age of invention) by Holland Thompson, with about 60,000 words, 2,500 sentences, and 6,800 distinct words,

6. *The adventures of Paddy the beaver* (paddy or Paddy), by Thornton W. Burgess, with about 18,000 words, 1,100 sentences, and 1,500 distinct words.

Two texts, "*The Two Captains*" and "*Collected tales by the Brothers Grimm*" are translations from their originals in German language into English.

For all six texts we sample the corresponding goSSR-, Bernoulli-, Simon-, and the random typewriting-model with the same sentence length distributions as the corresponding original text. Since the vocabulary of most of the texts is too large for computing the full word transition matrix for goSSR-, Simon-, and random typewriting-models, we restrict ourselves to word transitions between the 500 most frequent words in the texts. For notation, we use $K_{ij}$ for the number of times we observe the word $i$ to follow word $j$ in a sentence of a text. $A_{ij} = K_{ij}/k_j$ estimates the conditional probability for $i$ to follow $j$, where $k_j$ is the number of times word $j$ appears in the text.

We then compute two statistical measures that allow us to quantify properties of the transition matrices. The *skew* is a measure of the asymmetry of $A$, and the cos-measure is a proxy for the largest eigenvalue of $A$. We compute the transition matrices $A^{\text{text}}$, $A^{\text{bernoulli}}$, $A^{\text{goSSR}}$, $A^{\text{simon}}$, and $A^{\text{random}}$, and present the corresponding measures in Fig 3. The *top line* (first and third line of images) in Fig 3 shows the skew for all six texts, Moby Dick, Age of invention, Paddy, Two Captains, Five little peppers, and Grimm. The *bottom line* (second and fourth line) shows the cos-measure. The values corresponding to the original texts are shown as red lines. The values for the Bernoulli models (black) are averages over 5 realizations of the model. For the original text and the Bernoulli model the $x$-axis of the plot has no meaning since neither the original text nor the Bernoulli model depend on model parameters. For the random typewriting model (magenta) we fix the number of letters on the typewriter to $V = 30$, see Methods. For the goSSR and the Simon model the $x$-axis has a different meaning. For the goSSR model (blue) the $x$-axis is the number of grammatical labels $N_g = 1, \cdots, 12$, for the Simon model (green) it corresponds to the number of words, $W_0$ used for the initialization of the Simon model, see Methods. For all models we sample 5 realizations and present the mean and standard deviations. For the Bernoulli model the error bars are only slightly larger than the line-width. The goSSR model has been computed with a neutral element probability of $p_n = 0.1$ for the texts

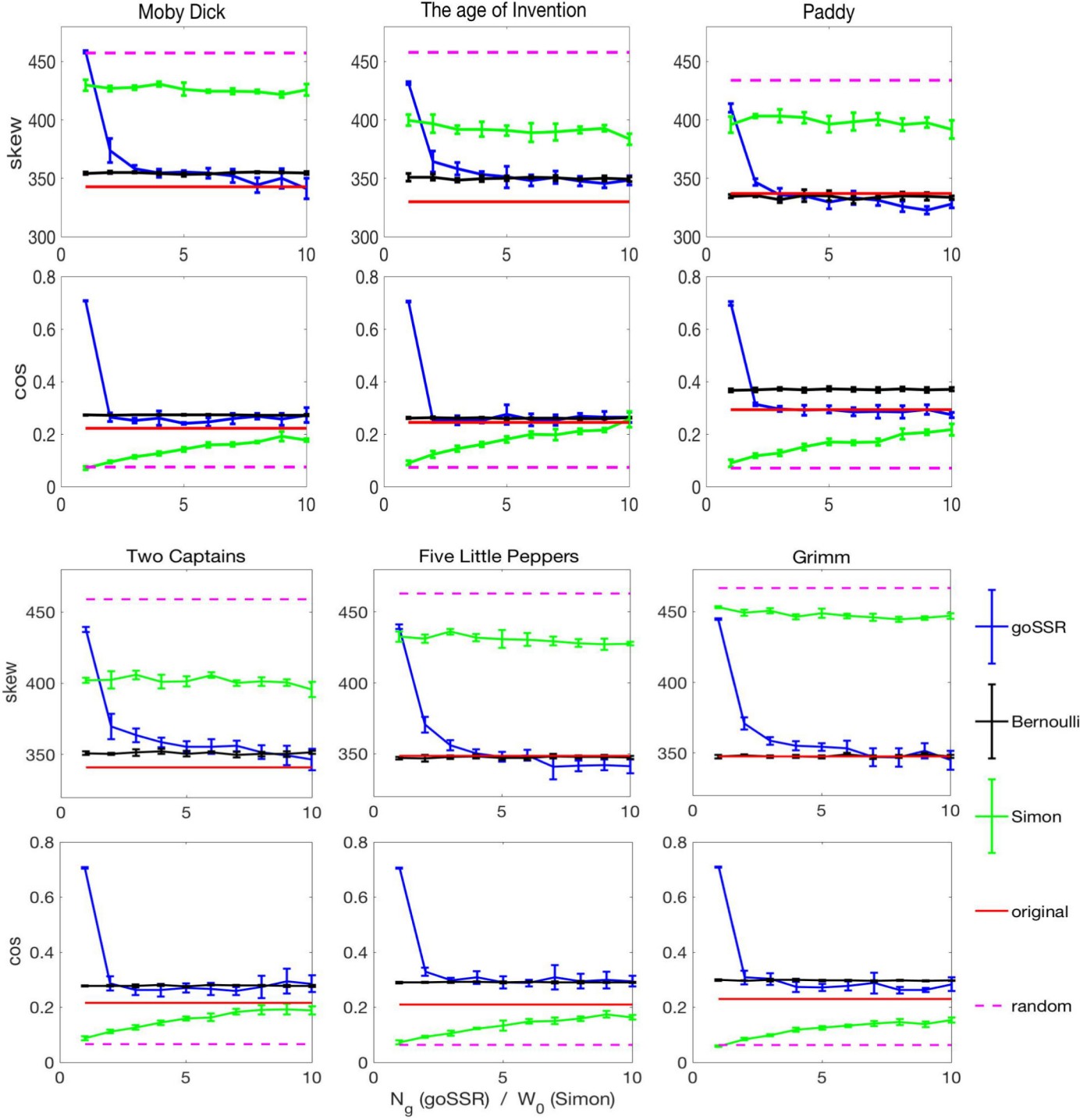

**Fig 3. Similarity measures.** For six texts in English language, two measures, the skew and a cos-measure, are used to compare the word transition matrices of the goSSR model (blue) with those of real texts (red), and their associated Bernoulli model (black). The values for the Simon model (green) with varying $W_0$ (from 1 to 10), and the random typewriting model (magenta) for $V = 30$ letters, are shown.

2cpns, 5lpep, and grimm. For moby, aoinvnt, and paddy results were obtained for $p_n = 0$. The value for the pure SSR models (no grammar) is given by $N_g = 1$.

For the six texts we observe in Fig 3 that for the two measures the goSSR model outperforms both, the Simon-, and the random-typewriting models. For increasing $N_g$ the goSSR model approaches both, the skew and the cos-measure of the real text. For levels of $N_g \sim 3 - 5$ convergence is typically reached. For the two translated texts, Two Captains, and Grimm, convergence takes slightly longer ($N_g \sim 6 - 9$).

For the skew, both, the random typewriting and Simon model can not explain the real text value. For the cos-measure the random typewriting model clearly fails. The Simon model approaches the real text value for large values of $W_0$. The cos-measure for the goSSR model is similar to the Bernoulli case, more so than the original text. There is almost no dependence on $N_g$, practically all values for $N_g > 1$ are similar. Since typically the skew of written text is close or identical to that of the Bernoulli model (see, 5lpep and grimm), one may be inclined to interpret the skew measure as something like the "grammatical depth" of a text. However, keep in mind that the Bernoulli model can per se not explain the Zipf law in word frequencies; that is a massive exogenous input.

In Table 1 we show the corresponding p-values. To compute those, consider the texts generated by the different models as data. We would like to test whether we can reject the null hypothesis that this data has been generated by the Bernoulli model at a confidence level of 0.05. To this end we sample the Bernoulli model of the particular text for 5, 000 times, compute the respective measure for each realization of the model, rank the values, and compute the respective p-values for the 5 realizations of each model. If a value is smaller than the smallest of the 5000 samples drawn for the test statistics, then we interpret the associated p-value as a number less than $1/5000 = 0.0002$. We present the average of the 5 p-values for the six texts. In Fig 3 we may observe that for both measures the original texts and their Bernoulli models are numerically similar, which confirms what one may expects from rough visual inspection of transition matrices of original text and corresponding Bernoulli models, compare Fig 2A and 2B. However, the p-values demonstrate that for a confidence level of 5% one would still have to reject the hypothesis that the original text is generated by its Bernoulli model for all six examples on the basis of the cos-measure and all but three case (5lpep, grimm, paddy) on the basis of the skew measure. The cases where we cannot reject the null hypothesis are for the Bernoulli model itself, which for all two test statistics is accepted with $p \sim 0.5$, and in most cases for the goSSR model for grammars with $N_g > 4$, see Table 1. For instance, the skew-measure of "Two Captains", for $N_g \geq 4$, p-values exceed the confidence level. That is, the transition probabilities of the goSSR process cannot be distinguished from the Bernoulli model with respect to the skew if the grammar becomes sufficiently complex. For two texts however, Mobby and Paddy, the cosine measure consistently rejects the null-hypothesis for the goSSR model and all values of the grammatical complexity $N_g$. Both measures indicate that original text in fact has transition matrices that typically resemble those of the corresponding Bernoulli model to a high degree, in terms of the value of measures, as we can see in Fig 3, but not to a degree that one could not be distinguished from the other on the basis of the considered similarity measures. For reasonable choices of $N_g$ and $p_n$, the goSSR model not only resembles the Bernoulli model closely but also accepts the null-hypothesis. The used measures indicate that the word transition matrices of goSSR models are located somewhere between real text and Bernoulli models in a statistical sense.

We considered additional measures to derive test-statistics that allow us to compare the rank-increment distributions of the real texts and the models. These included the $L_1$-norm, the Kolmogorov-Smirnov distance, and the Kullback-Leibler divergence. They are not shown here since they add little additional information. However, a note of caution for the naive

**Table 1. p-values for test statistics skew and cos.** The null hypothesis that the model texts have been generated by the Bernoulli model, at a 5% confidence level must be rejected if $p < 0.05$. Values for the goSSR model are shown for various $N_g$, the Simon model for $W_0 = 10$ and the random typewriting model for $V = 30$ words.

| | 2cpns | | 5lpep | | grimm | |
|---|---|---|---|---|---|---|
| | skew | cos | skew | cos | skew | cos |
| original | 0.0002 | 0.0002 | 0.59 | 0.0002 | 0.82 | 0.0002 |
| Bernoulli | 0.55 | 0.46 | 0.57 | 0.53 | 0.51 | 0.48 |
| $N_g = 1$ | 0.0002 | 0.0002 | 0.0002 | 0.0002 | 0.0002 | 0.0002 |
| $N_g = 3$ | 0.012 | 0.009 | 0.0002 | 0.0002 | 0.0002 | 0.02 |
| $N_g = 5$ | 0.18 | 0.07 | 0.21 | 0.12 | 0.025 | 0.0002 |
| $N_g = 7$ | 0.12 | 0.116 | 0.41 | 0.002 | 0.0002 | 0.0035 |
| $N_g = 10$ | 0.27 | 0.14 | 0.019 | 0.12 | 0.079 | 0.0002 |
| Simon | 0.0002 | 0.0002 | 0.0002 | 0.0002 | 0.0002 | 0.0002 |
| random | 0.0002 | 0.0002 | 0.0002 | 0.0002 | 0.0002 | 0.0002 |
| | moby | | aoinvnt | | paddy | |
| | skew | cos | skew | cos | skew | cos |
| original | 0.0002 | 0.0002 | 0.0002 | 0.0002 | 0.38 | 0.0002 |
| Bernoulli | 0.49 | 0.55 | 0.48 | 0.51 | 0.55 | 0.46 |
| $N_g = 1$ | 0.0002 | 0.0002 | 0.0002 | 0.0002 | 0.0002 | 0.0002 |
| $N_g = 3$ | 0.15 | 0.0002 | 0.14 | 0.038 | 0.23 | 0.0002 |
| $N_g = 5$ | 0.53 | 0.0002 | 0.02 | 0.0046 | 0.22 | 0.0002 |
| $N_g = 7$ | 0.29 | 0.0002 | 0.24 | 0.033 | 0.25 | 0.0002 |
| $N_g = 10$ | 0.07 | 0.0002 | 0.57 | 0.22 | 0.21 | 0.0002 |
| Simon | 0.0002 | 0.0002 | 0.0002 | 0.026 | 0.0002 | 0.0002 |
| random | 0.0002 | 0.0002 | 0.0002 | 0.0002 | 0.0002 | 0.0002 |

interpretation of p-values is necessary. The skew and cos-measure are among the most simple ones and are easy to interpret. There are many more, and it is conceivable that not all corresponding test-statistics necessarily confirm that the goSSR model outperforms the other models. In fact, we found that one test-statistic, based on the Kolmogorov-Smirnov rank-increment statistics, the Simon model sometimes performed slightly better than the corresponding goSSR model. What is true for all test-statistics we considered is that goSSR models show values of test-statistics that are typically located between real text and the Bernoulli model. Using different test-statistics for comparing models of complex phenomena may lead to distinct notions of similarity, which need not coincide. Different models may be adequate for some features (test-statistics) of the modelled phenomenon, while they may be quite inadequate with respect to others.

Following this line of reasoning one could continue by also augmenting other processes that generate Zipf's law with an additional grammatical ordering process. In this way one could go from a Simon model to a "goSimon" model, which now would be a two parameter model, depending on the initial vocabulary $W_0$ and the grammatical complexity $N_g$. Here, however, we contend ourselves with the observation that a grammatical ordering process that locally rearranges the word-order generated by a structured word selection process (here an SSR process) can easily explain why word-transition frequencies closely resemble those of a respective Bernoulli process, though–as we have learned here–typically not to the extent that word transition frequencies of original texts could not be distinguish from those of the respective Bernoulli model on the basis of suitable similarity measures.

## Discussion

The main objective of this paper was to demonstrate that a locally acting grammatical ordering process on top of a suitable word selection process, which we choose to be of SSR type, can explain both, the shape of the marginal distribution function, i.e. Zipf's law, and the random-looking shape of the empirical word transition distribution (the word transition matrix), as shown in Fig 2. Our results show that indeed, understanding the statistics of streams of English texts as a result of a grammatical ordering process (local reshuffling of words), superimposed on a SSR word selection process, is consistent with the evidence. While a pure SSR process offers a natural explanation for the observed (approximate) Zipf's laws in written texts, based on the necessity of contextual coherence, it fails to produce realistic word transition probabilities. Pure SSR transition probabilities are triangular and very different from empirical transition probabilities. The natural assumption that grammar is a process that locally rearranges word order, allows us to show that very simple grammatical rules are sufficient to explain the empirical structure of word transition matrices, in a statistical sense. Grammatical ordering that locally reshuffles selected word stream, in order to comply with grammatical structures, sufficiently destroys the triangular word transition structure of the SSR word selection process. goSRR models are therefore adequate, in the sense that they explain both the empirical word frequency distribution functions and basic statistical properties of the word transition probabilities of texts. One may note however, that augmenting other word selection processes generating Zipf's law (e.g. the Simon model) with a grammatical ordering process (e.g. to form a goSimon model) would also transform the respective word-transition matrices of those models to resemble those of a Bernoulli process more closely.

Of course, we can not say that actual English language is a SSR process augmented with a grammatical reordering process. We have seen that the goSSR model, at the level of statistics of the transition probabilities, performs similarly well as the Bernoulli model, which we know is a truly bad model. The Bernoulli model can not even explain the Zipf law in word frequencies. It can only explain features of the transition matrices, given that the Zipf law in word frequencies is provided as exogenous input. This paper demonstrated the possibility to consider word selection processes that generate Zipf's law and, by augmenting it with a grammatical ordering process that locally rearranges the word selection order, make it resemble a Bernoulli process in terms of its word-transition statistics.

Note that this particular *superposition* of processes exemplifies a more general phenomenon. If highly structured processes interact with each other, the resulting (very complex) process may look much more random than the underlying processes themselves. This emphasizes the often neglected fact, that statistical data alone is often insufficient for inferring the generative structure of the process that produces the data. Only if a specific parametric process class can be identified as being adequate for describing a given phenomenon, then data can be used to estimate which process within that class is likely to have generated the data. In other words, in the context of entangled, possibly *multi-causal* generative processes, even "big data" becomes worthless, without what is sometimes called a *thick description* of the phenomenon, which, in mathematical terms, is the requirement of having identified the process class that produces the observed phenomenon reasonably well. A minimal way to "thicken" a description consists of considering a spectrum of measures and models that provide clues to the underlying structure of a process instead of reducing a complex phenomenon to a singular model or notions of similarity.

## Methods

### Implementing grammatical word order

We first identify the full-stop, exclamation mark, and question mark in an original text as sentence ends, and obtain the sentence lengths in the text body. We produce a SSR sequence, $x = (x_1, \cdots, x_N)$, of $N$ words, using a vocabulary of $W$ words. The sequence is produced sentence by sentence, meaning that for every sentence in a text we generate a SSR sequence of the same length $m$ as the sentence $s = (s_1, \cdots, s_m) = (x_r, \cdots, x_{r+m-1})$, a sub-sequence of the text $x$, that starts at some position $r$ in the text. If the SSR process reaches word $i = 1$ in mid-sentence, the SSR process is continued by restarting the SSR process. Each $x_t$ takes integer values between 1 and $W$. For simplicity, assume here that there are no neutral words (grammatical label value 0). To every of the $W$ distinct words $i = 1, \cdots, W$, we randomly assign one of $N_g$ grammatical labels $L(i)$. We now work through $x$ sentence by sentence; the sentence length structure of the model is defined by the sentence lengths in the original text that we model. Let $s = (s_1, \cdots, s_m)$ be such a sentence. Then we form sub-sequences of $s$ consisting only of words with a particular grammatical label, $L(s_t) = \ell$, where $1 \leq t \leq m$, and $\ell = 1, \cdots, N_g$. Let $t_n(\ell)$ be the index of the $n$'th word in the sentence $s$ that carries the grammatical label $\ell$, then $s_\ell = (s_{t_1(\ell)}, s_{t_2(\ell)}, \ldots, s_{t_{n_\ell}(\ell)})$, where $n_\ell$ is the number of words with label $\ell$ in sentence $s$. The sequences $s_\ell$ are typically not of the same length. To make all sequences equally long we define $s_{t_n}(\ell)$ to be an "empty word", whenever $n > n_\ell$. In this way we can think of the sequences, $s_{t_n}(\ell)$, to be all of the same length $\hat{n} = \max\{n_\ell | 1 \leq \ell \leq N_g\}$ and parse them in lexicographical order with respect to $(t, \ell)$, where $(t', \ell') > (t, \ell)$, if $t' > t$, or $t' = t$ and $\ell' > \ell$.

The resulting sequence is the grammatically ordered sentence. Note that the grammatically ordered SSR sequence and the SSR sequence $x$ have identical word frequency distributions. If the SSR model explains Zipf's law, then so does the goSSR model. However, unlike SSR models, goSSR models now exhibit word transition probabilities that for a low enough fraction of neutral words (sufficiently small $p_n$), and a complex grammar (sufficiently large $N_g$) start to resemble those of real texts.

For example, with $N_g = 3$ classes, $1 \equiv S$, $2 \equiv P$, and $3 \equiv O$, for each sentence we get three sub-sequences, $s_S$, $s_P$, and $s_O$. $s_S = (s_{t_1(S)}, s_{t_2(S)}, \ldots, s_{t_{n_S}(S)})$ is the sub-sequence of all words in $s$ carrying label $S$. It contains all $n_S$ words of the sentence that carry the label $S$. We write $s_S(\tau) = s(t_\tau(S))$ for $\tau = 1, \cdots, n_S$. Similarly, $x_P$ and $x_O$ are the sub-sequences for labels $P$ and $O$, respectively. After following the procedure described above, we obtain the sequence

$$(s_S(1), s_P(1), s_O(1), s_S(2), s_P(2), s_O(2), \cdots) \quad . \tag{3}$$

Finally, by deleting the "empty words" we obtain a sentence that consists of the same words and has the same length as the SSR generated sentence $s$. This sentence is the grammatically ordered sentence.

If neutral words are present we proceed by combining neutral words with the next non-neutral word in a the sentence. For instance, if we find a sentence fragment of the form $\cdots, i, j, k, r, \cdots$ with grammatical labels $\cdots, 3, 0, 0, 2, \cdots$ then we consider $j, k, r$ as a single word $jkr$ with grammatical label $L(r) = 2$. After forming those *word blocks* we proceed as before with the word block inheriting the grammatical label of the non-neutral element it encapsulates.

### Word transitions of texts and models

How similar are word transition properties of the goSSR model to those of actual texts? For computational reasons we restrict the size of relative word transition frequency matrices $A$ to the $W_{\max} = 500$ most frequent words in a text. Since matrices for actual text, $A^{\text{text}}$, can not be

directly compared to those of a model, such as $A^{\text{goSRR}}$, we consider two appropriate statistical measures.

The first is a proxy for the largest eigenvalue of $A$, the cosine of the angle between the vector $v = (1, 1, \cdots, 1)$ and the vector $Av$,

$$\cos(A) = \frac{(v|Av)}{|v||Av|} \quad . \tag{4}$$

It measures how quickly the transition probabilities transform an equi-distributed set of words into the stationary empirical word frequency distribution.

The second is a measure for the asymmetry, the skew of $A$,

$$\text{skew}(A) \equiv \sum_{i=1}^{W_{\max}} \sum_{j=i}^{W_{\max}-1} (A_{ij} - A_{ji}) \quad . \tag{5}$$

## Bernoulli model

To keep the same sentence structure as in the original text, the Bernoulli model is obtained by first locating the positions of sentence-ends in the text. Then remove the sentence-ends and randomly re-shuffle the words of the entire text. Finally, we reinsert sentence ends at the previous positions in the text. We reshuffle the words of the text while keeping the length of the sentences fixed.

## Preferential Simon model

To generate a Simon model to fit a text with a vocabulary of $W$ words, and length $N$ we propose the following version of the Simon model. We initialize the process with a vocabulary of $W_0$ words with initial weights $k_i(t = 0) = k_0$, for $i = 1, \cdots, W_0$, and $k_i(t = 0) = 0$, for all other $i$. We use $k_0 = 1$. The probability $p_+(t)$ for sampling a new word at time step $t$ is computed from the size of the used vocabulary up to $t$, $W_{t-1}$, and the remaining number of time steps, $T + 1 - t$,

$$p_+(t) = \frac{W - W_{t-1}}{T + 1 - t} \quad .$$

The probability of sampling word $i$ at time $t$ is given by

$$p_i(t) = (1 - p_+(t)) \frac{k_i(t)}{\sum_{j=1}^{W} k_j(t)} \quad .$$

Every time a word $i$ is sampled at time $t$, we increase $k_i(t + 1) = k_i(t) + 1$. In this way the process follows a Simon type of update rule, while adapting its parameters to match length and vocabulary of a given text.

## Random typewriting model

Assume we have a keyboard (alphabet) with $V$ letters and a space key. The probability to hit the space key is $p_w(t)$. We initialize the model with an empty lexicon. For each time step $t = 1, \cdots, N$, we sample a random sequence of letters (words), where each letter is produced with probability $(1 - p_w(t))/V$. If a space is hit, sampling letters stops, and it is checked, if the random word already exists in the lexicon. If not, the word is added to the lexicon, and its word-

count, $k_{word}$, is set to one; otherwise, the word count is increased by one. Knowing the number of words sampled up to time $t$, $W(t)$, and the number of time steps $T + 1 - t$ to go, one can adjust $p_w$ to control the total number of words sampled by random typewriting. Heuristic considerations suggest for example to choose $p_w(t) = 1/(1 + q(t))$, with $q(t) = \log((W - W(t) + 1)(V - 1) + 1)/\log(V) - 1$. For the random typewriting process text length and vocabulary size are harder to be matched with real texts.

## Acknowledgments

We gladly acknowledge important and insightful discussions on the presented topic with our dear colleague Bernat Cormominas-Murtra.

## Author Contributions

**Conceptualization:** Rudolf Hanel, Stefan Thurner.

**Data curation:** Rudolf Hanel.

**Formal analysis:** Rudolf Hanel.

**Funding acquisition:** Stefan Thurner.

**Methodology:** Rudolf Hanel.

**Project administration:** Stefan Thurner.

**Validation:** Rudolf Hanel, Stefan Thurner.

**Writing – original draft:** Rudolf Hanel.

**Writing – review & editing:** Rudolf Hanel, Stefan Thurner.

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
