## [Decision Letter · Decision Letter 0]

23 Jul 2020

PONE-D-20-13476

The role of grammar in transition-probabilities of subsequentwords in English text

PLOS ONE

Dear Dr. Hanel,

Thank you for submitting your manuscript to PLOS ONE. After careful consideration, we feel that it has merit but does not fully meet PLOS ONE’s publication criteria as it currently stands. Therefore, we invite you to submit a revised version of the manuscript that addresses the points raised during the review process.

Please accept our apologies for the tardiness in providing you with feedback. The reason being to wait for a second opinion of a steamed colleague that eventually could not do it. We hope that you find the reviewer' comments useful enough to turn around a revised version of your manuscript. 

We look forward to receiving your revised manuscript.

Kind regards,

Dante R. Chialvo

Academic Editor

PLOS ONE

Journal Requirements:

Reviewers' comments:

Reviewer's Responses to Questions

**Comments to the Author**

1. Is the manuscript technically sound, and do the data support the conclusions?

Reviewer #1: Partly

2. Has the statistical analysis been performed appropriately and rigorously? 

Reviewer #1: Yes

3. Have the authors made all data underlying the findings in their manuscript fully available?

Reviewer #1: Yes

4. Is the manuscript presented in an intelligible fashion and written in standard English?

Reviewer #1: No

5. Review Comments to the Author

Reviewer #1: In this manuscript, the authors present a statistical analysis of a model for text generation which combines sample-space reduction (SSR) and a kind of stylized "grammatical" order. They show that a handful of simple but sensible rules are able to reproduce statistical properties of real texts and, not unexpectedly, conclude that SSR-like processes and grammatical constraints are at the basis of word usage and ordering in written laguange.

Overall, the paper is technically correct and presents interesting and original results, so it should eventually be published. I see however three main drawbacks in presentation and methodology, that the authors should deeply revise for resubmission.

1. At several points, the text is unacceptably imprecise. It sometimes makes the impression that the authors didn't give it a thorough reading before submission. For instance:

a. We are told that Fig. 1 shows data for "The Two Captains", but the figure caption says otherwise.

b. In section 0.1 (?!), we read

"Imagine the first word in a sentence is randomly drawn from the entire lexicon with W words (states), say "The wolf". As soon as it is decided, the second word must be a verb".

First, "The wolf" are two words, not one. Second, it is not true that grammar imposes that the next word is a verb. I can imagine infinitely many sentences where the next word is "of", "in", "that", "however", "therefore", etc. etc. etc.

c. The labels S, P and O, used at various points in the text, are never defined.

d. At the beginning of section 1, we are told that the analysis concerns three novels. However, "Five Little Peppers" is rather a series of novels, and the "Collected Tales by the Brothers Grimm" is -quite obviously- a collection of tales, not a novel.

These are just four examples of a long series of presentation inconsistencies that the authors must carefully consider and solve in a revision.

2. In the statistical analysis of language, it is not good practice to work with translated texts. The effects of translation on the statistics of word usage, vocabulary choice, grammatical correlations, and so on, have never been assessed, and depend upon a series of more or less uncontrolled factors. Just think of the quality of translation. Statistical measurements on translated texts might be biased by elements whose characterization would need a fully separate -not yet undertaken- study. Unfortunately, of a total of three works, the present results concern two translations. I urge the authors to redo their analysis for texts in their original language. It would also help interpreting the results if the works have similar lengths and vocabulary sizes.

3. A major drawback of the model of text generation presented in this manuscript is that the SSR process acts at the level of sentences only. Namely, the process is renewed at the beginning of each sentence. However -as has been discussed at length in connection with Simon's model for Zipf's law- the effect of word choices extends over much longer scales, at the level of full chapters and even the entire work. The reason for this is, of course, evident: the initial choice of words establishes the semantic context of the full work -concretely, what the work tells us about- and therefore strongly constrains the appearance of new words over the whole text.

This point is completely overlooked in the manuscript. By itself, such failure does not invalidate the contribution. However, the point is so obvious that it is mandatory, for the sake of scientific correctness, that the authors comment on it in sufficient detail. It would be welcome, in fact, that they discuss how the model could be extended to encompass longer contextual spans.

In summary, I cannot recommend publication in the present form, but I acknowledge that the manuscript can be improved in several ways to make it acceptable for PLOS ONE.

6. PLOS authors have the option to publish the peer review history of their article (what does this mean?). If published, this will include your full peer review and any attached files.

Reviewer #1: No

---

## [Author Response · Author response to Decision Letter 0]

8 Sep 2020

Dear reviewer, we thank you for the efforts you have taken to review our manuscript and point out mistakes and points worth of improvement. A detailed response to your comments are attached in the Cover Letter of the manuscript. 

With best regards

Rudolf Hanel

---

## [Decision Letter · Decision Letter 1]

18 Sep 2020

The role of grammar in transition-probabilities of subsequent

words in English text

PONE-D-20-13476R1

Dear Dr. Hanel,

We’re pleased to inform you that your manuscript has been judged scientifically suitable for publication and will be formally accepted for publication once it meets all outstanding technical requirements.

Kind regards,

Dante R. Chialvo

Academic Editor

PLOS ONE

Additional Editor Comments (optional):

Reviewers' comments:

Reviewer's Responses to Questions

**Comments to the Author**

1. If the authors have adequately addressed your comments raised in a previous round of review and you feel that this manuscript is now acceptable for publication, you may indicate that here to bypass the “Comments to the Author” section, enter your conflict of interest statement in the “Confidential to Editor” section, and submit your "Accept" recommendation.

Reviewer #1: (No Response)

2. Is the manuscript technically sound, and do the data support the conclusions?

Reviewer #1: Yes

3. Has the statistical analysis been performed appropriately and rigorously? 

Reviewer #1: Yes

4. Have the authors made all data underlying the findings in their manuscript fully available?

Reviewer #1: Yes

5. Is the manuscript presented in an intelligible fashion and written in standard English?

Reviewer #1: Yes

6. Review Comments to the Author

Reviewer #1: The authors have reasonably addressed all my criticisms, and the paper has been improved in several directions. I can now recommend its publication in PLOS ONE.

7. PLOS authors have the option to publish the peer review history of their article (what does this mean?). If published, this will include your full peer review and any attached files.

Reviewer #1: **Yes: **Damián H. Zanette

---

## [Editor Report · Acceptance letter]

25 Sep 2020

PONE-D-20-13476R1 

The role of grammar in transition-probabilities of subsequent
words in English text 

Dear Dr. Hanel:

I'm pleased to inform you that your manuscript has been deemed suitable for publication in PLOS ONE. Congratulations! Your manuscript is now with our production department. 

Kind regards, 

on behalf of

Dr Dante R. Chialvo 

Academic Editor

PLOS ONE